# Hemodialysis Treatment for Patients with Lithium Poisoning

**DOI:** 10.3390/ijerph191610044

**Published:** 2022-08-15

**Authors:** Yu-Hsin Liu, Kai-Fan Tsai, Pai-Chin Hsu, Meng-Hsuan Hsieh, Jen-Fen Fu, I-Kuan Wang, Shou-Hsuan Liu, Cheng-Hao Weng, Wen-Hung Huang, Ching-Wei Hsu, Tzung-Hai Yen

**Affiliations:** 1Department of Anaesthesiology, Chang Gung Memorial Hospital, Linkou, Taoyuan 333, Taiwan; 2Division of Nephrology, Department of Internal Medicine, Kaohsiung Chang Gung Memorial Hospital and Chang Gung University College of Medicine, Kaohsiung 833, Taiwan; 3Division of Nephrology, Department of Internal Medicine, Taoyuan General Hospital, Ministry of Health and Welfare, Taoyuan 330, Taiwan; 4Department of Medical Research, Chang Gung Memorial Hospital, Linkou, Taoyuan 333, Taiwan; 5College of Medicine, Chang Gung University, Taoyuan 333, Taiwan; 6Department of Nephrology, China Medical University Hospital, Taichung 406, Taiwan; 7Department of Nephrology, Clinical Poison Center, Chang Gung Memorial Hospital, Linkou, Taoyuan 333, Taiwan

**Keywords:** lithium, poisoning, psychiatric comorbidity, hemodialysis, acute kidney injury network score, mortality

## Abstract

Background: Hemodialysis is often recommended to treat severe lithium poisoning. Nevertheless, the application rate of hemodialysis in patients with lithium poisoning is varied across different groups and the effect of hemodialysis is still undetermined. Therefore, this study aimed to analyze the hemodialysis rate of patients with lithium poisoning and to explore the clinical features of lithium-poisoned-patients treated or untreated with hemodialysis. Methods: Between 2001 and 2019, 36 patients treated at the Chang Gung Memorial Hospital for the management of lithium poisoning were stratified according to whether they were treated with hemodialysis (*n* = 7) or not (*n* = 29). Results: The patients were aged 50.7 ± 18.1 years. The poisoning patterns were acute on chronic (61.1%), chronic (25.0%) and acute (13.9%). The precipitating factors of dehydration and infection were noted in 36.1% and 25.0% of patients, respectively. Bipolar disorder (72.2%), depressive disorder (27.8%) and psychotic disorder (11.1%) were the top three psychiatric comorbidities. The hemodialysis group not only had a lower Glasgow Coma Scale (GCS) score (*p* = 0.001) but also had a higher respiratory failure rate (*p* = 0.033), aspiration pneumonia rate (*p* = 0.033) and acute kidney injury network (AKIN) score (*p* = 0.002) than the non-hemodialysis group. Although none of the patients died of lithium poisoning, the hemodialysis group required more endotracheal intubation (*p* = 0.033), more intensive care unit admission (*p* = 0.033) and longer hospitalization (*p* = 0.007) than the non-hemodialysis group. Conclusion: The analytical results revealed zero mortality rate and low hemodialysis rate (1.9%). Compared with patients without hemodialysis, patients receiving hemodialysis suffered severer lithium-associated complications and needed a more intensive care unit admission and longer hospital stay.

## 1. Introduction

Lithium salt is a popular medication for bipolar disorder, but it has a narrow therapeutic index (0.6–1.2 mEq/L), and lithium toxicity may occur in cases of excessive intake or decreased excretion [1]. Lithium poisoning had a low mortality rate (Table 1) [1,2,3,4,5,6,7,8,9,10,11,12,13,14,15,16,17,18,19]. Although one study [7] reported a mortality rate of 12.5%, most studies have reported low mortality rates. One group [5] reported a zero mortality rate in their study, even though the studied patients had a mean blood lithium level of 3.43 mEq/L. The published mortality rates of patients with lithium poisoning were 0–12.5% before 2010. After 2011, the mortality rates dropped to 0–4.3%. Notably, when the sample size was larger than 100, the mortality rate was between 0.2% and 3.1%.

Hemodialysis is an important procedure for treating patients with severe lithium poisoning because it enhances lithium clearance. As shown in Table 1 [1,2,3,4,5,6,7,8,9,10,11,12,13,14,15,16,17,18,19], the published hemodialysis rates for lithium poisoning vary between 1.5% and 60.8%. However, the indications and benefits of hemodialysis remain controversial [20]. In 2015, the EXTRIP Workgroup published a systematic review of the indications for hemodialysis treatment for lithium poisoning [21]. According to these guidelines, hemodialysis is an ideal extracorporeal treatment for patients with lithium poisoning and the EXTRIP Workgroup supported the use of extracorporeal treatment for severe lithium poisoning. Nevertheless, clinical decisions on when to use extracorporeal treatment should consider the blood lithium level, renal function, pattern of lithium toxicity, patient condition and availability of hemodialysis services. Therefore, whether the indications for hemodialysis suit all types of lithium poisoning is still a matter of debate [17,18].

Lithium is not metabolized and is eliminated via the kidneys. Plasma lithium concentrations are sensitive to physiological factors that influence kidney function, for instance age, fluid status, hemodynamics and sodium balance [22,23]. Therefore, drug interactions do not include biotransformation through enzyme systems such as cytochrome P450. On the other hand, the drug interactions occur through the direct effect of other medicines upon renal function and glomerular filtration rate. Thiazide, loop diuretics, angiotensin-converting enzyme inhibitors, angiotensin receptor blockers and non-steroidal anti-inflammatory drugs were known for their influence on increasing plasma lithium level. Some other medicines such as antidepressants, antipsychotics, antiepileptic drugs and non-depolarizing neuromuscular blockers do not alter the lithium serum level, but may cause adverse neurotoxicity.

Hemodialysis is often recommended to treat severe lithium poisoning. Nevertheless, the application rate of hemodialysis in patients with lithium poisoning is varied across different groups and the effect of hemodialysis is still undetermined. Given the recommendation by the EXTRIP Workgroup of the use of hemodialysis treatment for severe lithium poisoning, we hypothesized that hemodialysis treatment could exert clinical improvement for patients with severe lithium poisoning by providing rapid extracorporeal removal of lithium. Therefore, the objective of this study was to analyze the hemodialysis rate of patients with lithium poisoning and to explore the clinical features of lithium-poisoned-patients treated or untreated with hemodialysis.

## 2. Materials and Methods

### 2.1. Patients

Between 2001 and 2019, 36 patients were treated at Chang Gung Memorial Hospital for the management of lithium poisoning. Medical records, including emergency department records, admission, progress and discharge records, nursing records, psychiatric consultation records and outpatient clinic records, were reviewed. Demographic data, such as age, sex, marital status and employment status, were collected, along with the patients’ medical and psychiatric comorbidities. Information on lithium therapy, such as the dose and duration, was also obtained. The patterns of poisoning, precipitating factors and clinical findings were studied. Clinical, physical and laboratory findings, including complete blood counts, biochemistry and lithium levels, were analyzed. Clinical outcomes were assessed and treatment modalities were evaluated.

### 2.2. Patient Group

Patients with lithium poisoning were stratified according to whether they were treated with hemodialysis (*n* = 7) or not (*n* = 29).

### 2.3. Inclusion and Exclusion Criteria

All patients aged ≥ 18 years with confirmed lithium poisoning were recruited for the analysis. Patients aged < 18 years were excluded from the study. Patients were also excluded if they did not have detectable lithium levels in the blood despite suspicion of lithium poisoning.

### 2.4. Diagnosis of Lithium Poisoning

Lithium poisoning was diagnosed on the basis of exposure history, clinical findings and physical examinations, and the poisoning was confirmed by a blood lithium test. Blood lithium levels were determined using a spectrophotometric method. The therapeutic level was 0.6–1.2 mEq/L, warning level was 1.2–1.5 mEq/L and toxic level was >1.5 mEq/L.

### 2.5. Clinical Event Definitions

The clinical patterns of lithium poisoning were defined as: acute poisoning, arising in patients not previously receiving lithium; acute-on-chronic poisoning, resulting from acute ingestion during lithium therapy; and chronic poisoning, therapeutic misadventure due to insufficient monitoring, unsuitably high target levels or intercurrent diseases [24]. The Glasgow Coma Scale (GCS) was scored according to ocular, verbal and motor responses. The lowest score was 3 (deep coma), whereas the highest score was 15 (completely awake). Acute kidney injury was scored according to the Acute Kidney Injury Network (AKIN) criteria [25], which defines and categorizes the severity of acute kidney injury in three stages using serum creatinine and urine output.

### 2.6. Detoxification Protocol

The goal of detoxification therapy was to ascertain lithium concentration < 1 mEq/L stably within 6 h from hemodialysis. Patients were treated with large amounts of normal saline, as fluid resuscitation is crucial in the initial management of lithium poisoning. The protocol did not include forced emesis, polyethylene glycol enema or sodium polystyrene sulfonate resin. The decision of performing hemodialysis depended on serial lithium levels, renal function and clinical condition.

### 2.7. Indications for Hemodialysis Treatment

The indications for hemodialysis included [24] (1) any patient with a lithium level > 6 mEq/L; (2) any patient on chronic lithium therapy with a lithium level > 4 mEq/L; (3) any patient with serum lithium levels ranging from 2.5 to 4.0 mEq/L, with severe neurologic symptoms, renal failure or hemodynamic instability; (4) any patient with a serum lithium level < 2.5 mEq/L with end-stage kidney disease; and (5) any patient whose lithium levels rose after admission or failed to reach a lithium level < 1 mEq/L in 30 h.

### 2.8. Hemodialysis Procedure

The procedure was performed for 4 h using a femoral vein catheter [1]. The blood and dialysate flow rates were 200 and 500 mL/min, respectively. Zero ultrafiltration volume was prescribed to avoid fluid removal. Hemodialysis was conducted using hollow-fiber dialyzers fitted with modified cellulose membranes. The dialysate used had a standard ionic composition with bicarbonate buffer and a standard reverse osmosis system technique was employed for water purification.

### 2.9. Statistical Analysis

All data were tested for the normality of distribution and equality of standard deviations before analysis. Continuous variables are expressed as mean and standard deviations and categorical variables are presented as numbers and percentages in parentheses. Student’s *t*-test was used to analyze quantitative variables and the chi-square or Fisher’s exact test was used for categorical variables. A *p*-value of less than 0.05 was selected as the significance threshold to reject the null hypothesis. All analyses were performed using the IBM SPSS Statistics version 20.0 (IBM Corp., New York, NY, USA).

## 3. Results

Table 2 shows the baseline characteristics of patients with lithium poisoning, stratified according to whether they were treated with hemodialysis or not. Four patients received one treatment of hemodialysis, and the other three patients underwent hemodialysis treatment twice. The patients had an average age of 50.7 ± 18.1 years and 44% of the patients were male. Hypertension and diabetes mellitus were noted in 30.6% and 22.2% of patients, respectively. Most of the poisonings were acute on chronic (61.1%), followed by chronic (25.0%) and acute (13.9%). The mean daily dose of lithium was 675.0 ± 260.8 mg, and the mean duration of lithium therapy was 42.6 ± 69.1 months. Regarding the precipitating factors for lithium poisoning, dehydration and infection were found in 36.1% and 25.0% of patients, respectively. There were no significant differences in the baseline data between both groups.

As shown in Table 3, bipolar disorders (72.2%), depressive disorders (27.8%) and psychotic disorders (11.1%) were the top three psychiatric comorbidities among patients with lithium poisoning. There were no significant differences in psychiatric comorbidities between the groups.

Neurological complications were the predominant symptoms: 33.3% of patients had muscle tremors, 30.6% had drowsiness and 19.4% had dizziness, ataxia or muscle weakness (Table 4). The hemodialysis group not only had a lower GCS score (*p* = 0.001), but also had a higher respiratory failure rate (*p* = 0.033), higher aspiration pneumonia rate (*p* = 0.033) and higher AKIN score (*p* = 0.002) than the non-hemodialysis group. Three patients suffered renal diabetes insipidus, with average urine outputs and specific gravity of 6880 mL, 4750 mL and 6900 mL urine per day, as well as 1.005, 1.005 and 1.003, respectively.

The hemodialysis group had higher peak serum creatinine levels than the non-hemodialysis group (*p* = 0.002, Table 5). In addition, the hemodialysis group also had higher blood lithium levels than the non-hemodialysis group (3.2 ± 12 versus 2.5 ± 1.0), but this was not statistically significant (*p* = 0.135). There was no evidence of new onset diabetes and no changes in diabetic treatment after lithium poisoning.

As shown in Table 6, none of the patients died of lithium poisoning. The hemodialysis group required more endotracheal intubation and mechanical ventilation (*p* = 0.033), more intensive care unit admission (*p* = 0.033) and longer hospitalization (*p* = 0.007) than the non-hemodialysis group.

## 4. Discussion

The hemodialysis group had a higher AKIN score than the non-hemodialysis group (*p* = 0.002; Table 5). Acute kidney injury (AKI) is common in patients with lithium poisoning. According to a systematic review by the EXTRIP Workgroup [21], acute kidney injury occurred in 32.3% of patients with acute and acute-on-chronic lithium poisoning and 58.3% of patients with chronic lithium poisoning. In this study, acute kidney injury was observed in 47.3% of the patients (Table 5). The pathophysiological mechanism underlying the effects of lithium on decreased glomerular filtration rate remains poorly understood. Once lithium is ingested, it is distributed throughout the body within a few hours. Lithium is not metabolized and is excreted entirely in urine. Approximately, 80% of lithium is filtered by the glomerulus and is reabsorbed: 60% by the proximal tubule and 20% by the thick ascending limb of the loop of Henle and collecting duct [21]. A number of renal dysfunctions related to lithium have been reported [26], including tubular defects, which may result in nephrogenic diabetes insipidus and glomerular defects, which may lead to a decrease in the glomerular filtration rate.

Two patients (5.6%; Table 6) experienced respiratory failure and required mechanical ventilator support. Both the patients had acute-on-chronic poisoning. Both patients had a low Glasgow Coma Scale (6 and 4, respectively) on hospital arrival. One patient showed labored breath and the other experienced apnea during hemodialysis. Aspiration pneumonia was found in two cases. According to a literature analysis, the mechanical ventilation rates of patients with lithium poisoning ranged between 1.7% and 13.6% [8,10,13]. Respiratory failure is an uncommon manifestation of lithium poisoning; however, it can be induced by consciousness disturbance, severe neurological symptoms, aspiration pneumonia or acute respiratory distress syndrome (ARDS). Five cases of lithium-associated ARDS have been reported in the literature [27,28,29,30]. These five patients had blood lithium levels of 3.3–4.9 mEq/L. Other causes of ARDS such as cardiogenic pulmonary edema, neurogenic pulmonary edema, pulmonary embolism, diffuse pneumonia, toxin injury and dialysis-induced hypoxia were excluded. According to Bloomfield and Young [31], lithium can enhance neutrophil degranulation and induce inflammation in the pulmonary parenchyma in animal models. In addition, Weiner et al. [32] stated that lithium can depress the brain respiratory drive.

Most patients had psychiatric comorbidities such as bipolar disorder (72.2%) or depressive disorder (27.8%, Table 3). Four patients had psychotic disorders, three patients had alcoholism and one patient had dementia. Lithium therapy is indicated mainly in patients with bipolar disorder and sometimes in depressed patients [33,34]. The mechanisms of action of lithium therapy are only partially understood. According to Alda [35], lithium may reduce excessive neuronal activity and contribute to the stabilization of neuronal activity, stress resilience, neuronal or synaptic plasticity and regulation of chronobiological function. In other studies [12,16], schizophrenia and schizoaffective disorder were psychiatric comorbidities that may also use lithium as an augmentation of antipsychotic medication, and these patients could also experience lithium poisoning. Alcohol abuse is another possible psychiatric comorbidity reported by Lopez et al. [12]. Therefore, patients with psychiatric comorbidities may have access to lithium and, at the same time, they have a higher risk of impulse usage or overdosing themselves with medication. Therefore, understanding psychiatric comorbidities may help clinicians evaluate their clinical condition and prevent the incidence of lithium poisoning.

As shown in Table 2, acute chronic poisoning (61.1%) was the primary cause of poisoning. Moreover, acute chronic poisoning comprised 85.7% of the cases in the hemodialysis group. In the studies that analyzed three types of poisoning patterns [1,3,6,7,8,10,18,19], acute-on-chronic patterns ranged from 24.7% to 84.9%. In studies that analyzed acute on chronic as the predominant pattern [1,3,5,6,12,17], the proportions of acute-on-chronic poisoning were 47.6–84.9%. Additionally, higher blood lithium levels have been reported in patients with acute-on-chronic poisoning than in those with chronic poisoning [18].

This study showed a zero-mortality rate and low hemodialysis rate (1.9%). The favorable outcome was similar to data from other international poison centers. As shown in Table 1 [1,2,3,4,5,6,7,8,9,10,11,12,13,14,15,16,17,18,19], the published mortality and hemodialysis rates of patients with lithium poisoning were 0–12.5% and 1.5–60.8%, respectively. Most published lithium studies are from Western countries, but a few studies are from Asian countries. Accessibility to medical care and hemodialysis intervention is higher in Western countries than in Asian countries. Therefore, it would be helpful for clinicians to understand the outcomes of patients with lithium poisoning by collecting epidemiological data from Asian countries. Finally, this study was limited by its retrospective design, small patient population, considerable variation in the age of the patient population and the small number of hemodialysis procedures performed. Therefore, further studies are warranted.

## 5. Conclusions

The analytical results revealed a zero-mortality rate and low hemodialysis rate (1.9%). The poisoning patterns were acute on chronic (61.1%), followed by chronic (25.0%) and acute (13.9%). Bipolar disorders (72.2%), depressive disorders (27.8%) and psychotic disorders (11.1%) were the top three psychiatric comorbidities. Importantly, patients receiving hemodialysis had lower GCS scores, higher AKIN scores and higher rates of respiratory failure, and therefore required more intensive care unit admission (*p* = 0.033) and longer hospitalization periods than patients without hemodialysis.

## Figures and Tables

**Table 1 ijerph-19-10044-t001:** Literature study on the outcomes of hemodialysis treatment for patients with lithium poisoning.

Time	Study	Area	Sample Size	Peak Blood Lithium Level (mEq/L)	Hemodialysis Rate	Mortality Rate
2022	Current study	Taiwan	36	2.64	1.9%	0%
2020	Buckley et al. [18]	Australia	361	1.85	2.5%	0.2%
2020	Vodovar et al. [17]	France	128	3.2	17.2%	3.1%
2020	Hlaing et al. [16]	Australia	22	2.2	13.6%	0%
2020	Chan et al. [15]	Australia	242	Not available	1.6%	0.4%
2016	Ott et al. [19]	Sweden	91	Not available	13.2%	4.3%
2016	Vodovar et al. [14]	France	128	3.2	17.2%	3.1%
2013	Bretaudeau Deguigne et al. [13]	France	59	3.65	18.6%	3.4%
2012	Lopez et al. [12]	Spain	65	Not available	21.0%	0%
2011	Dennison et al. [11]	Ireland	47	2.62	11.0%	Not available
2011	Lee et al. [1]	Taiwan	21	2.54	14.2%	0%
2010	Offerman et al. [10]	USA	502	2.92	13.7%	0.8%
2010	Ghannoum et al. [9]	Canada	48	1.91	14.5%	0%
2006	Eyer et al. [8]	German	22	2.77	40.9%	4.5%
2002	Meltzer and Steinlauf [7]	Israel	8	Not available	Not available	12.5%
2000	Bailey and McGuigan [6]	Canada	205	2.10	Not available	1%
1993	Jaeger et al. [5]	France	14	3.43	21.4%	0%
1988	Gadallah et al. [4]	United States	55	Not available	9.1%	0%
1987	Dyson et al. [3]	United Kingdom	68	Not available	1.5%	Not available
1978	Hansen and Amdisen [2]	Denmark	23	3.2	60.8%	8.6%

**Table 2 ijerph-19-10044-t002:** Baseline characteristics of patients with lithium poisoning, stratified according to whether they were treated with hemodialysis or not (*n* = 36).

Variables	Hemodialysis Patients (*n* = 7)	Non-Hemodialysis Patients (*n* = 29)	All Patients (*n* = 36)	*p*-Value
Age, year	48.7 ± 15.7	51.2 ± 18.8	50.7 ± 18.1	0.766
Male, *n* (%)	2 (28.6)	14 (48.3)	16 (44.4)	0.426
Body mass index, kg/m^2^	24.2 ± 2.7	26.2 ± 3.9	25.7 ± 3.7	0.297
Unemployed, *n* (%)	6 (85.7)	21 (72.4)	27 (75.0)	0.652
Living alone, *n* (%)	0 (0)	3 (10.3)	3 (8.3)	1.000
Pattern of poisoning, *n* (%)				0.291
Acute, *n* (%)	0 (0)	5 (17.2)	5 (13.9)	
Acute on chronic, *n* (%)	6 (85.7)	16 (55.2)	22 (61.1)	
Chronic, *n* (%)	1 (14.3)	8 (27.6)	9 (25.0)	
Lithium dose, mg/day	480.0 ± 268.3	732.4 ± 236.5	675.0 ± 260.8	0.055
Duration of lithium therapy, month	6.4 ± 8.2	49.1 ± 73.2	42.6 ± 69.1	0.208
Precipitating dehydration, *n* (%)	2 (28.6)	11 (37.9)	13 (36.1)	1.000
Precipitating infection, *n* (%)	1 (14.3)	8 (27.6)	9 (25.0)	0.652
Hypertension, *n* (%)	3 (42.9)	8 (27.6)	11 (30.6)	0.650
Diabetes mellitus, *n* (%)	2 (28.6)	6 (20.7)	8 (22.2)	0.639
Cardiovascular disease, *n* (%)	0 (0)	2 (6.9)	2 (5.6)	1.000
Cerebrovascular, *n* (%)	0 (0)	1 (3.4)	1 (2.8)	1.000
Thyroid disease, *n* (%)	1 (14.3)	0 (0)	1 (2.8)	0.194
Hepatic disease, *n* (%)	0 (0)	3 (10.3)	3 (8.3)	1.000
Kidney disease, *n* (%)	0 (0)	3 (10.3)	3 (8.3)	1.000
Neurological disease, *n* (%)	2 (28.6)	2 (6.9)	4 (11.1)	0.163
Gastrointestinal disease, *n* (%)	0 (0)	2 (6.9)	2 (5.6)	1.000
Malignancy, *n* (%)	0 (0)	1 (3.4)	1 (2.8)	1.000
Smoking habit, *n* (%)	1 (14.3)	10 (34.5)	11 (30.6)	0.400
Alcohol consumption, *n* (%)	1 (14.3)	8 (27.6)	9 (25)	0.625
Medications				
Thiazide, *n* (%)	0 (0)	1 (3.4)	1 (2.8)	1.000
Loop diuretics, *n* (%)	0 (0)	3 (10.3)	3 (8.3)	1.000
Angiotensin-converting enzyme inhibitors, *n* (%)	0 (0)	0 (0)	0 (0)	1.000
Angiotensin receptor blockers, *n* (%)	2 (28.6)	2 (6.9)	4 (11.1)	0.163
Non-steroidal anti-inflammatory drugs, *n* (%)	1 (14.3)	0 (0)	1 (2.8)	0.194

**Table 3 ijerph-19-10044-t003:** Psychiatric comorbidities of patients with lithium poisoning, stratified according to whether they were treated with hemodialysis or not (*n* = 36).

Psychiatric Diagnosis	Hemodialysis Patient (*n* = 7)	Non-Hemodialysis Patients (*n* = 29)	All Patients (*n* = 36)	*p*-Value
Adjustment disorder, *n* (%)	0 (0)	0 (0)	0 (0)	1.000
Depressive disorder, *n* (%)	2 (28.6)	8 (27.6)	10 (27.8)	1.000
Bipolar disorder, *n* (%)	5 (71.4)	21 (72.4)	26 (72.2)	1.000
Psychotic disorder, *n* (%)	1 (14.3)	3 (10.3)	4 (11.1)	1.000
Substance abuse, *n* (%)	0 (0)	0 (0)	0 (0)	1.000
Alcoholic, *n* (%)	0 (0)	3 (10.3)	3 (8.3)	1.000
Delirium or dementia, *n* (%)	0 (0)	1 (3.4)	1 (2.8)	1.000

**Table 4 ijerph-19-10044-t004:** Clinical manifestations of patients with lithium poisoning, stratified according to whether they were treated with hemodialysis or not (*n* = 36).

Variables	Hemodialysis Patient (*n* = 7)	Non-Hemodialysis Patients (*n* = 29)	All Patients (*n* = 36)	*p*-Value
Body temperature, °C	37.0 ± 0.7	36.8 ± 0.6	36.9 ± 0.6	0.450
Heart rate, per minute	100.0 ± 30.8	79.4 ± 19.6	83.4 ± 23.2	0.033 *
Respiratory rate, per minute	19.0 ± 2.6	18.8 ± 1.9	18.8 ± 2.0	0.813
Systolic blood pressure, mmHg	126.0 ± 14.9	126.7 ± 26.5	126.5 ± 24.5	0.950
Diastolic blood pressure, mmHg	71.4 ± 14.3	76.3 ± 15.3	75.4 ± 15.0	0.448
Glasgow Coma Scale score	10.9 ± 4.3	14.2 ± 1.3	13.6 ± 2.5	0.001 **
Neurological complications				
Drowsiness, *n* (%)	4 (57.1)	7 (24.1)	11 (30.6)	0.167
Slurred speech, *n* (%)	1 (14.3)	4 (13.8)	5 (13.9)	1.000
Dizziness, *n* (%)	1 (14.3)	6 (20.7)	7 (19.4)	1.000
Ataxia, *n* (%)	1 (14.3)	6 (20.7)	7 (19.4)	1.000
Muscle twitching, *n* (%)	2 (28.6)	1 (3.4)	3 (8.3)	0.090
Muscle weakness, *n* (%)	1 (14.3)	6 (20.7)	7 (19.4)	1.000
Rigidity, *n* (%)	2 (28.6)	4 (13.8)	6 (16.7)	0.573
Tremor, *n* (%)	3 (42.9)	9 (31.0)	12 (33.3)	0.664
Cardiovascular complications				
Hypertension, *n* (%)	0 (0)	1 (3.4)	1 (2.8)	1.000
Hypotension, *n* (%)	0 (0)	1 (3.4)	1 (2.8)	1.000
Sinus tachycardia, *n* (%)	1 (14.3)	2 (6.9)	3 (8.3)	0.488
Sinus bradycardia, *n* (%)	1 (14.3)	4 (13.8)	5 (13.9)	1.000
Corrected QT interval prolongation, *n* (%)	2 (28.6)	1 (3.4)	3 (8.3)	0.090
Respiratory complication				
Respiratory failure, *n* (%)	2 (28.6)	0 (0)	2 (5.6)	0.033 *
Aspiration pneumonia, *n* (%)	2 (28.6)	0 (0)	2 (5.6)	0.033 *
Gastrointestinal complications				
Nausea and vomiting, *n* (%)	2 (28.6)	10 (34.5)	12 (33.3)	1.000
Anorexia, *n* (%)	0 (0)	7 (24.1)	7 (19.4)	0.303
Diarrhea, *n* (%)	4 (57.1)	6 (20.7)	10 (27.8)	0.076
Renal complications				
AKIN score				0.002 **
Class 1	0 (0)	10 (34.5)	10 (27.8)	
Class 2	0 (0)	2 (6.9)	2 (5.6)	
Class 3	4 (57.1)	1 (3.4)	5 (13.9)	
Total	4 (57.1)	13 (44.8)	17 (47.2)	
Diabetes insipidus, *n* (%)	1 (14.3)	2 (6.9)	3 (8.3)	0.488

Note: AKIN acute kidney injury network; * *p* < 0.05, ** *p* < 0.01.

**Table 5 ijerph-19-10044-t005:** Laboratory data of patients with lithium poisoning, stratified according to whether they were treated with hemodialysis or not (*n* = 36).

Variables	Hemodialysis Patient (*n* = 7)	Non-Hemodialysis Patients (*n* = 29)	All Patients (*n* = 36)	*p*-Value
White blood cell count, 1000/µL	12,957.1 ± 4919.6	11,427.6 ± 5472.8	11,725.0 ± 5337.3	0.504
Neutrophils, %	75.7 ± 11.6	73.9 ± 12.1	74.3 ± 11.9	0.728
Hemoglobin, g/dL	12.0 ± 1.4	12.5 ± 1.2	12.4 ± 1.2	0.347
Mean corpuscular volume, fL	89.4 ± 4.4	92.4 ± 5.1	91.8 ± 5.1	0.162
Red cell distribution width, %	13.8 ± 1.2	13.2 ± 0.6	13.3 ± 0.8	0.052
Platelet count, 1000/µL	277.1 ± 77.9	246.8 ± 76.9	252.7 ± 77.0	0.357
Alanine aminotransferase, U/L	53.1 ± 73.6	33.7 ± 41.6	38.0 ± 49.5	0.368
Random glucose, mg/dL	141.0 ± 77.2	123.9 ± 39.6	127.8 ± 48.8	0.506
Random glucose, mg/dL (after discharge)	160.0 ± 71.2	152.0 ± 38.1	155.1 ± 47.9	0.832
Blood urea nitrogen, mg/dL	39.6 ± 27.5	24.5 ± 17.7	27.3 ± 20.1	0.134
Creatinine, mg/dL	3.6 ± 3.0	1.5 ± 0.8	1.9 ± 1.7	0.002 **
Potassium, mEq/L	3.7 ± 0.9	4.1 ± 0.6	4.0 ± 0.7	0.155
Sodium, mEq/L	137.0 ± 4.7	139.0 ± 4.0	138.6 ± 4.1	0.248
Calcium	8.8 ± 0.4	9.1 ± 1.0	9.0 ± 0.8	0.439
Thyroid stimulating hormone	2.3 ± 2.4	2.9 ± 1.7	2.8 ± 1.8	0.554
Lithium (arrival), mEq/L	3.2 ± 12	2.5 ± 1.0	2.6 ± 1.1	0.135
Lithium (after hemodialysis), mEq/L	1.2 ± 0.7	1.0 ± 0.5	1.0 ± 0.6	0.326
Interval between first and second lithium test, hour	38.4 ± 25.6	78.2 ± 100.4	70.5 ± 91.8	0.311

Note: ** *p* < 0.01.

**Table 6 ijerph-19-10044-t006:** Outcomes of patients with lithium poisoning, stratified according to whether they were treated with hemodialysis or not (*n* = 36).

Variables	Hemodialysis Patient (*n* = 7)	Non-Hemodialysis Patients (*n* = 29)	All Patient (*n* = 36)	*p*-Value
Hospitalization duration, day	16.9 ± 8.9	8.1 ± 6.7	9.9 ± 7.9	0.007 **
Intensive care unit admission, *n* (%)	2 (28.6)	0 (0)	2 (5.6)	0.033 *
Endotracheal intubation and mechanical ventilation, *n* (%)	2 (28.6)	0 (0)	2 (5.6)	0.033 *
Mortality, *n* (%)	0 (0)	0 (0)	0 (0)	1.000

Note: * *p* < 0.05, ** *p* < 0.01.

## Data Availability

The datasets used and analyzed for this study are available from the corresponding author upon request.

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
