# Peer review of "Hemodialysis Treatment for Patients with Lithium Poisoning"

_ijerph, 2022, doi:10.3390/ijerph191610044_

Round 1
Reviewer 1 Report
Thank you for your changes.
Nothing further from me.
Reviewer 2 Report
I believe that the presented manuscript can be accepted for publication.
Reviewer 3 Report
Nope
This manuscript is a resubmission of an earlier submission. The following is a list of the peer review reports and author responses from that submission.
Round 1
Reviewer 1 Report
A small number of samples
Response. Thank you for the comment. We understand that small sample size comprise a limitation of this study. The limitation has been addressed in the Discussion paragraph.
Reviewer 2 Report
A clear goal of the study - whether to lower the lithium concentration or achieve clinical improvement.
Response. Thank you for the comment.
There is no information in the introduction about pharmacological interactions with lithium. There is no information on the use of other drugs that affect the pharmacokinetics of lithium.
Response. Thank you for the comment. The Introduction paragraph has been expanded.
Lithium is not metabolized and is eliminated via kidney. Plasma lithium concentrations are sensitive to physiological factors that influencing kidney function for instance age, fluid status, hemodynamics and sodium balance [22, 23]. Therefore, drug interactions do not include biotransformation through enzyme systems such as cytochrome P450. On the other hand, the drug interactions occur through the direct effect of other medications upon renal function and glomerular filtration rate. Thiazide, loop diuretics, angiotensinconverting enzyme inhibitors, angiotensin receptor blockers and non-steroidal anti-inflammatory drugs were known for their influence on increasing plasma lithium level. Some other medications like antidepressants, antipsychotics, antiepileptic drugs, non-depolarizing neuromuscular blockers do not alter lithium serum level, but may cause adverse neurotoxicity.
The use of thiazide, loop diuretics, angiotensin-converting enzyme inhibitors, angiotensin receptor blockers and non-steroidal anti-inflammatory drugs have been provided in Table 2.
Please better define the assumption and purpose of the work.
Response. Thank you for reminding us. The hypothesis and objective of the work has been revised.
Hemodialysis is often recommended to treat severe lithium poisoning. Nevertheless, the application rate of hemodialysis in patients with lithium poisoning is varied across different groups and the effect of hemodialysis is still undetermined. Given the recommendation by EXTRIP Workgroup of use of hemodialysis treatment for severe lithium poisoning, we hypothesized that hemodialysis treatment could exert clinical improvement for patients with severe lithium poisoning by providing rapid extracorporeal removal of lithium. Therefore, this study analyzed the hemodialysis rate of patients with lithium poisoning, and explored the clinical features of lithium poisoned-patients treated or untreated with hemodialysis.
It is advisable to provide information on whether the electrolyte levels were assessed - the concentration of glucose, sodium, potassium, TSH, calcium, information on the amount of urine output and its density (renal diabetes insipidus).
Response. Thank you for the comment. The data of glucose, sodium, potassium, thyroid stimulating hormone and calcium has been provided in Table 5.
Three patients suffered renal diabetes insipidus, with average urine outputs and specific gravity of 6880 mL, 4750 mL and 6900 mL urine per day, as well as 1.005, 1.005 and 1.003, respectively.
It is advisable to provide at least the lithium concentration after hemodialysis and how many dialysis treatments were performed, what was the target lithium concentration.
Response. Thank you for the comment. The lithium concentrations on arrival (first) and after hemodialysis treatment (second) have been provided in Table 5. Four patients received one time of hemodialysis, and the other three patients underwent hemodialysis treatment for twice. The procedure was performed for 4 hours using a femoral vein catheter. The goal of hemodialysis therapy was to ascertain lithium concentration < 1 mEq/L stably within 6 hours from hemodialysis.
How was HD conducted - ultrafiltration rates, HD fluid, were patients diuresis?
Response. Thank you for the comment. All patients received one time of hemodialysis, and the procedure was performed for 4 hours using a femoral vein catheter [1]. The blood and dialysate flow rates were 200 and 500 mL/min, respectively. Zero ultrafiltration volume was prescribed to avoid fluid removal. Hemodialysis was conducted using hollow-fiber dialyzers fitted with modified cellulose membranes. The dialysate used had a standard ionic composition with bicarbonate buffer, and a standard reverse osmosis system technique was employed for water purification.
Goal or goal achieved, which should be lithium concentration <1 mEq / L stably within 6 hours from HD.
Response. Thank you for reminding us. This important target has been stressed.
Has sodium polystyrene sulfonate been used?
Response. Thank you for the comment. The detoxification protocol did not include forced emesis, polyethylene glycol enema or sodium polystyrene sulfonate resin.
The age of the patient population varied considerably, which resulted in a change in the volume of distribution of lithium.
Response. Thank you for the comment. This limitation has been included in the Discussion paragraph.
Finally, this study was limited by its retrospective design, small patient population, considerable variation in age of the patient population, and small number of hemodialysis procedures performed.
22.2% of patients had diabetes - how lithium influenced glycemia and whether diabetes treatment needed to be modified.
Response. Thank you for the comment. There was no evidence of new onset diabetes and no changes in diabetic treatment after lithium poisoning.
Discussion: AKI in 47.3% of patients - the table does not show this.
Response. Thank you for the comment. The total number of patients with AKI has been included in Table 4.
The applications do not correspond to the title of the work purpose.
Response. Thank you for the comment. The hypothesis and objective of the work has been revised. Therefore, the article seems more correspond to the title of the work purpose. Please kindly advise if we missed out anything.
Hemodialysis is often recommended to treat severe lithium poisoning. Nevertheless, the application rate of hemodialysis in patients with lithium poisoning is varied across different groups and the effect of hemodialysis is still
undetermined. Given the recommendation by EXTRIP Workgroup of use of hemodialysis treatment for severe lithium poisoning, we hypothesized that hemodialysis treatment could exert clinical improvement for patients with severe lithium poisoning by providing rapid extracorporeal removal of lithium. Therefore, the objective of this study was to analyze the hemodialysis rate of patients with lithium poisoning, and to explore the clinical features of lithium
poisoned-patients treated or untreated with hemodialysis.
Reviewer 3 Report
This paper examines lithium poisoning and management of that by hemodialysis (HD).
Unfortunately, in their cohort of 36 patients, only 7 received HD. It is a retrospective cohort study.
Response. Thank you for the comment. We understand that small sample size comprise a limitation of this study. The limitation has been addressed in the Discussion paragraph.
The paper does not really help us sort out who should or should not be managed with HD. The abstract and conclusion state that those that received HD had a few worse parameters and longer hospital stays than those not treated by HD. But which is cause and which is effect. Presumably worse patients (higher levels) received HD and thus represented sicker patients - who will therefore have worse parameters.
Response. Thank you for the comments.
Hemodialysis is often recommended to treat severe lithium poisoning. Nevertheless, the application rate of hemodialysis in patients with lithium poisoning is varied across different groups and the effect of hemodialysis is still
undetermined. As shown in Table 1 [1-19], the published hemodialysis rates for lithium poisoning vary between 1.5% and 60.8%. Given the recommendation by EXTRIP Workgroup of use of hemodialysis treatment for severe lithium
poisoning, we hypothesized that hemodialysis treatment could exert clinical improvement for patients with severe lithium poisoning by providing rapid extracorporeal removal of lithium. Therefore, the objective of this study was to
analyze the hemodialysis rate of patients with lithium poisoning, and to explore the clinical features of lithium poisoned-patients treated or untreated with hemodialysis.
What we need to know is:
- what was the decision process around deciding to use HD?
Response. Thank you for the comment. The indications for hemodialysis included [24] (1) any patient with a lithium level > 6 mEq/L; (2) any patient on chronic lithium therapy with a lithium level > 4 mEq/L; (3) any patient with serum lithium levels ranging from 2.5 to 4.0 mEq/L, with severe neurologic symptoms, renal failure, or hemodynamic instability; (4) any patient with serum lithium level < 2.5 mEq/L with end-stage kidney disease, any patient whose lithium levels rose after admission, or failed to reach a lithium level < 1 mEq/L in 30 hours.
- what was the trajectory of improvement in Li levels for those with or without HD management?
Response. Thank you for the comment. The lithium concentrations on arrival (first) and after hemodialysis treatment (second) have been provided in Table 5.
- apart from length of hospital stay, we don't have a feel for the outcomes (apart from low mortality).
Response. Thank you for the comment. Apart from length of hospital stay, it was found that the hemodialysis group required more endotracheal intubation and mechanical ventilation (P = 0.033) and more intensive care unit admission
(P = 0.033) than the non-hemodialysis group.
There are quite a few superfluous data points. Does it help to know what season of the year it was? There is no discussion as to why this was entered and no discussion on its influence on outcome.
Response. Thank you for the comment. As instructed, the season of the year variable has been removed.
Most patients had psychiatric comorbidities such as bipolar disorder (72.2%) or depressive disorder (27.8%, Table 3). Four patients had psychotic disorders, three patients had alcoholism, and one patient had dementia. Lithium therapy is indicated mainly in patients with bipolar disorder and sometimes in depressed patients [33, 34]. The mechanisms of action of lithium therapy are only partially understood. According to Alda [35], lithium may reduce excessive neuronal activity and contribute to the stabilization of neuronal activity, stress resilience, neuronal or synaptic plasticity, and regulation of chronobiological function. In other studies [12, 16], schizophrenia and schizoaffective disorder were psychiatric comorbidities that may also use lithium as an augmentation of antipsychotic medication, and these patients could also experience lithium poisoning. Alcohol abuse is another possible psychiatric comorbidity reported by Lopez et al [12]. Therefore, patients with psychiatric comorbidities may have access to lithium and, at the same time, they have a higher risk of impulse usage or overdosing themselves with medication. Therefore, understanding psychiatric comorbidities may help clinicians evaluate their clinical condition and prevent the incidence of lithium poisoning.
Similarly, I don't think the marital status helps. Much of the psycho-social data is less relevant here. It would be OK to include this in a Psychiatry Journal paper but given the focus here is on usefulness and indications for HD, it could be reduced.
Response. Thank you for the comment. As instructed, the marital status variable has been removed.